# Profiles and interactions of gut microbiome and intestinal microRNAs in pediatric Crohn's disease

Yao Lv,[1] Changjun Zhen,[1] Ana Liu,[1] Yudie Hu,[1] Gan Yang,[1] Cuifang Xu,[1] Yue Lou,[1] Qi Cheng,[1] Youyou Luo,[1] Jindan Yu,[1] Youhong Fang,[1] Hong Zhao,[1] Kerong Peng,[1] Yu Yu,[1] Jingan Lou,[1] Jie Chen,[1] Yan Ni[1]

**ABSTRACT** Gut dysbiosis is closely related to dysregulated microRNAs (miRNAs) in the intestinal epithelial cells, which plays an important role in the pathogenesis of Crohn's disease (CD). We investigated the relationship between fecal gut microbiome (GM) and intestinal tissue miRNAs in different stages of pediatric CD. Metagenomic analysis and miRNA sequencing were conducted to examine the GM and intestinal miRNA profiles of CD patients before and after clinical induction therapy and the controls. Twenty-seven newly diagnosed, therapy-naïve pediatric patients with active CD and 11 non-inflammatory bowel disease (IBD) controls were recruited in this study. Among CD patients, 11 patients completed induction treatment and reached clinical remission. Both GM and miRNA profiles were significantly changed between CD patients and controls. Seven key bacteria were identified at species level including *Defluviitalea raffinosedens*, *Thermotalea metallivorans*, *Roseburia intestinalis*, *Dorea* sp. AGR2135, *Escherichia coli*, *Shigella sonnei*, and *Salmonella enterica*, the exact proportions of which were further validated by real-time quantitative PCR analysis. Eight key miRNAs were also identified including hsa-miR-215-5p, hsa-miR-194-5p, hsa-miR-12135, hsa-miR-509-3-5p, hsa-miR-212-5p, hsa-miR-4448, hsa-miR-501-3p, and hsa-miR-503-5p. The functional enrichment analysis of differential miRNAs indicated the significantly altered cyclin protein, cyclin-dependent protein, and cell cycle pathway. The close interactions between seven key bacteria and eight key miRNAs were further investigated by miRNA target prediction. The association between specific miRNA expressions and key gut bacteria at different stages of CD supported their important roles as potential molecular biomarkers. Understanding the relationship between them will help us to explore the molecular mechanisms of CD.

**IMPORTANCE** Since previous studies have focused on the change of the fecal gut microbiome and intestinal tissue miRNA in pediatric Crohn's disease (CD), the relationship between them in different stages is still not clear. This is the first study to explore the gut microbiota and miRNA and their correlations with the Pediatric Crohn's Disease Activity Index (PCDAI). Crohn's Disease Endoscopic Index of Severity (CDEIS), and calprotectin, by applying two omics approach in three different groups (active CD, CD in remission with exclusive enteral nutrition or infliximab induction therapy, and the healthy controls). Both gut microbiome structure and the miRNA profiles were significantly changed in the different stage of CD. Seven key gut microbiome at species and eight key miRNAs were found, and their close interactions were further fully investigated by miRNA target prediction.

**KEYWORDS** gut microbiome, miRNAs, pediatric Crohn's disease, interactions, clinical induction therapy

C rohn's disease (CD) is a chronic inflammatory bowel disease (IBD) with increasing incidence worldwide (1). Although the precise etiology of CD remains elusive,

Address correspondence to Yan Ni, yanni617@zju.edu.cn, or Jie Chen, 6185020@zju.edu.cn.

The authors declare no conflict of interest.

recent studies have indicated a multifaceted interplay between genetic susceptibility, environmental factors, immunological anomalies, and perturbations in gut microbiota (2). Among those, the intestinal microbiota plays an important role in the pathogenesis of CD. Patients with CD exhibit less diverse microbial communities compared with healthy counterparts characterized by a decrease in *Bacteroides* and *Firmicutes* bacteria, as supported by our prior investigation and corroborated by other studies (3, 4). Animal studies have confirmed that disrupted gut microbial balance contributed to the onset and progression of colitis (5). Compelling evidence also endorses fecal microbiota transplantation as an effective therapy in CD (6).

The intestinal microbiota can both be affected by both intrinsic host factors such as microRNAs (miRNAs) and mucus barrier and external factors including diet (7). MiRNAs, a class of small non-coding RNAs, approximately 18–22 nucleotides long, undergo nuclear and cytoplasmic processing and exert regulatory control over targeted gene expression (8). Within the gut, host miRNAs, which are predominantly secreted by intestinal epithelial cells and then released into the lumen, critically impact CD pathophysiology by modulating the growth and compositions of gut microbiota (9). Several miRNAs, including miR-101-5p, miR-325-3p, miR-515-5p, miR-1224-5p, miR-1226-5p, and miR-1253-5p, have been identified to infiltrate bacteria like *Fusobacterium nucleatum* and *Escherichia coli*, influencing bacterial gene expression and growth and consequently shaping the compositions and distribution of the intestinal microbiota (9). Studies performed by Johnston et al. (10) confirmed that miR-21-5p can influence dextran sodium sulfate-induced colitis by reshaping the gut microbiota. Distinct fecal miRNA expressions in IBD patients, including miR-199a, miR-1226-5p, miR-548a, and miR-515-5p, have been demonstrated to influence the growth of bacteria like *E. coli*, *F. nucleatum*, and *segmented filamentous bacteria* (11).

In turn, gut microbiota in the host intestine can regulate the miRNA expression. Xue et al. (12) showed that gut microbiota downregulated miR-107 in intestinal CD11c$^+$ myeloid cells. Nine differential miRNAs were found in the ileum and colon from mice colonized with specific microbiota compared to germ-free mice (13). Another study has highlighted that 16 miRNAs were differentially expressed in the cecum of conventional male mice relative to germ-free mice (14). Furthermore, intestinal microbiota can also affect miRNA expressions through the metabolites they produce. For example, butyrate, produced by intestinal microbiota, can downregulate Prdm1 and Aicda in human and mouse B cells by upregulating miRNAs such as miR-26a, miR-30c, and miR-200c (15). Dietary lipids including conjugated linoleic acid and docosahexanoic acid could impact the expression of miR-107 in Caco-2 cells (16).

In this study, we applied two omics approach to characterize the structure and functions of the gut microbiome (GM) via fecal samples and miRNA expressions from mucosal tissue samples in three different groups (active CD [aCD], CD in remission, and the healthy controls). Meanwhile, we explored their associations between GM and miRNA profiles and calculated their relationship with clinical key indicators such as Pediatric Crohn's Disease Activity Index (PCDAI), Crohn's Disease Endoscopic Index of Severity (CDEIS), and calprotectin. At last, we investigated the key miRNAs and predicted gut microbiota gene targets. This study aims to delineate the intricate miRNA–microbiota interactions in CD, shedding light on the underlying mechanisms and providing comprehensive insights into this subject.

## MATERIALS AND METHODS

### Study design

Newly diagnosed, therapy-naïve patients with CD according to ESPGHAN Revised Porto Criteria (17) based on endoscopy, biopsies, clinical manifestations, and/or radiological findings and age- and gender-matched no-IBD patients were prospectively enrolled in the study from September 2019 to August 2022 in the Children's Hospital, Zhejiang University School of Medicine, a major referral center for children with IBD in China. The

inclusion criteria for patients with CD were defined as follows: (i) aged from 6 years to 18 years, PCDAI ≥10 points when diagnosed as CD at the first time and (ii) who achieved PCDAI <10 points after induction therapy. The following were excluded for non-IBD control (Ctr) criteria: (i) who had the family history of IBD, (ii) who used antibiotics or probiotics within 2 months before sample collection, and (iii) who had the inflamed mucosa in the colon. Patients who achieved the PCDAI ≥10 points before induction therapy were defined as aCD, and patients with PCDAI <10 points after exclusive enteral nutrition (EEN) or infliximab (IFX) induction therapy were classified as remission CD (rCD). Those rCDs before induction therapy were defined as aCDs (aCD subgroup).

## Sample collection

Stool samples of patients with CD including aCD and rCD and Ctr patients were collected in sterile containers and then homogenized immediately and stored in −80°C freezer until analysis. Two punch biopsies of mucosal tissue from terminal ileum were collected from each subject. In aCD group the colonoscopy punch biopsies were obtained from inflamed mucosa, and in rCD and Ctr groups, the colonoscopy punch biopsies were collected from healing mucosa and uninflamed mucosa, respectively. Tissue samples were stored in RNAlater in −80°C freezer immediately.

## Isolation of mucosal tissue RNA and miRNA library construction

Total RNA from mucosal tissue was extracted using TRIzol (Life Technologies Corp, USA), further ligates adapters to each end of the RNA molecule, and then reverse transcribes and amplifies to produce a cDNA library. A gel purification step prepares the library for clustering and sequencing. cDNAs were then treated with DNase to remove genomic DNA contamination, and the rest of DNase would be inactivated later. Isolation of mRNA was performed using the NEBNext PolyA mRNA Magnetic Isolation Module (New England Biolabs, Ipswich, MA, USA), and the mRNA was then used for RNA-sequencing library preparation with the NEB Next Ultra Directional RNA Library Prep Kit for Illumina (New England Biolabs, Ipswich, MA, USA) following the manufacturer's instructions.

## miRNA sequencing and quality control

Pair-end 2 × 150 mode of the Illumina Novaseq 6000 platform was used as sequencing mode. Raw reads were filtered to obtain high-quality clean reads by removing sequencing adapters, short reads (length <15 bp) and low-quality reads using Cutadapt v1.18 (18) (non-default parameters: --max-n 0 --minimum-length 15 –M 35) and Trimmomatic v0.38 (19) (non-default parameters: SLIDINGWINDOW:4:15 LEADING:10 TRAILING:10 MINLEN:15). Then FastQC (20) (with default parameters) is used for ensure high reads quality. Then FastQC (3) (with default parameters) is used to ensure high reads quality. The resulting clean reads were mapped to human mature miRNA genome (assembly Rnor_6.0) using the Bowtie v1.2.2 (21) non-default parameters: -v 1 k 1 m 1 --best--strata) software. Using miRBase as the reference (http://www.mirbase.org), reads of each miRNA were determined, and reads per million was adopted as the normalization method to quantify the abundance of each miRNA.

## Analysis of differential miRNA expression

Differential miRNA expression analyses were performed in DESeq2 v1.22.2 (22). The false discovery rate control method was used to calculate the adjusted $P$ values in multiple testing in order to measure the significance of the differences. Here, only genes with an adjusted $P < 0.05$ and $|\log_2$ fold change (FC)$| > 1$ were used for subsequent analysis.

## Functional enrichment analysis based on eight key miRNAs

The target genes of hsa-miR-215-5p, hsa-miR-194-5p, hsa-miR-12135, hsa-miR-509-3-5p, hsa-miR-212-5p, hsa-miR-4448, hsa-miR-501-3p, and hsa-miR-503-5p were predicted on

miranda, PITA databases, and RNAhybrid. The gene ontology (GO) and Kyoto Encyclopedia of Genes and Genomes (KEGG) pathway enrichment analyses were performed on clusterProfiler package of R (23). The GO or KEGG pathway terms with $P < 0.05$ were considered significant.

## Metagenomic sequencing and analysis

Fecal DNA was extracted using DP712 Kit (Tiangen, China) following manufacturer's instructions. All DNA samples were used for library construction by NEBNext Ultra DNA Library Prep Kit for Illumina (NEB, USA). Paired-end metagenomic sequencing was performed on an Illumina Novaseq 6000 platform (24). The clean data acquired by Trimmomatic (0.39) was conducted on the Illumina HiSeq sequencing platform (25) using Readfq (V8) and then assembled and analyzed on the SOAPdenovo software (V2.04) (26). The fragment shorter than 500 bp in all of Scaftigs generated from single or mixed assembly were excluded in our analysis.

### Gene prediction and abundance analysis

Open reading frame prediction was conducted on MetaGeneMark (version 2.10) (27). The unique initial gene catalog was obtained from CD-HIT software (version 4.5.8) (28). Clean reads were then mapped to the initial gene catalog using Bowtie2.2.4 (29). The abundance information of each gene in each sample according to the number of mapped reads and the length of gene was then analyzed. All the basic information statistics, core-pan gene analysis, correlation analysis of samples, and Venn figure analysis of number of genes were based on the abundance of each gene in each sample in gene catalog.

### Taxonomy prediction

The unigenes to the sequences of bacteria, fungi, archaea, and viruses extracted from the Non-Redundant Protein Sequence Database (NR database; version 2018-01-02) of National Center for Biotechnology Information was blasted on the DIAMOND (30) software (version 0.9.9). The abundance information and number of genes of each sample in each taxonomy hierarchy (kingdom, phylum, class, order, family, genus, and species) were aquired according to the gene abundance table and the non-redundant unigenes annotation result.

### Function enrichment analysis

Linear discriminant analysis (LDA) effect size was conducted to compare the differences of KEGG pathways of microbial community compositions among groups. Only LDA scores ($\log_{10}$) > 2 and $P < 0.05$ were considered significantly enriched.

## RT-PCR for bacteria detection

Fecal DNA extraction was described as above, and equal amounts (~10 ng/µL) of cDNA template were used for real-time PCR (RT-PCR). The following primers (forward primer, f-ACTCCTACGGGAGGCAGCAG, and reverse primer, r-ATTACCGCGGCTGCTGG) were determined as internal control (31), and a-no DNA reaction was conducted as negative control. Specific primers for *Defluviitalea raffinosedens*, *Roseburia intestinalis*, *Dorea* sp. AGR2135, *Thermotalea metallivorans*, *E. coli*, *Shigella sonnei*, and *Salmonella enterica* were the following: f-TAACCTACCTTACACAGGGGGATA, r-TCTCACCAACTAGCTAATCAGACG; f-AGAGTTTGATCCTGGCTCAGGATG, r-TACTCACCCGTCCGCCAC; f-AAACCCCACTCGAAGCAAGAC, r-AAAGCAAGCAAGACGATAAGCCG; f-GCGTAGATATTAGGAGGAACACCA, r-CGTGGACTACCAGGGTATCTAATC; f-ACCTTCGGGCCTCTTGC, r-GTCTCAGTTCCAGTGTGGCT; f-ATGCGTAGAGATCTGGAGGAATAC, r-TACCAGGGTATCTAATCCTGTTTG; f-ACGGTAGCTAATACCGCATAATGT, r-GTTACCTCACCAACAAGCTAATCC. The PCR conditions were set as follows: 50°C for 2 min, 95°C for 10 min, 40 cycles of 95°C for 15 s, 56°C for 30 s, 72°C for 60 s, and a final

72°C for 10 min. Amplifications were performed using the SYBR Green (Vazyme, China) and a RT-quantitative PCR (RT-qPCR) system (Bio-Rad, USA).

## miRNA target prediction

The seed sequence of hsa-miR-215-5p, hsa-miR-194-5p, hsa-miR-12135, hsa-miR-509-3-5p, hsa-miR-212-5p, hsa-miR-4448, hsa-miR-501-3p, and hsa-miR-503-5p were BLAST-searched against the *D. raffinosedens*, *R. intestinalis*, *Dorea* sp. AGR2135, *T. metallivorans*, *E. coli*, *S. sonnei*, and *S. enterica* genome for sequence pairing using the blastn v2.13.0.

## Statistical analysis

Data analysis was performed in R software 3.6.3. The Mann–Whitney *U* test and Student's *t*-test were used to examine the differences of clinical parameters and miRNA as continuous variables. The Fisher's exact test was performed to compare categorical variables. FC was calculated to evaluate the median difference of fecal microbiota between two groups (e.g., Ctr vs aCD). The GM analysis and its association analysis with miRNAs were conducted using M$^2$IA pipeline (32). For gut microbiota diversity measures, we investigated associations for Shannon index (33). β-Diversity derived with multivariate principal coordinate analysis (PCoA) was used to investigate the microbial community composition (34). Spearman's correlation analysis was performed to examine the correlations between miRNAs/GM and PCDAI, CDEIS score, and calprotectin. The correlation between miRNAs and GM was present in Sankey charts (35) with Spearman's rank correlations. $P < 0.05$ was considered statistically significant. $P < 0.01$ was considered highly significant.

## RESULTS

### Clinical features of patients with CD and Ctr

The workflow and process of participant enrollment were present in Fig. 1. Eleven of 27 reached clinical remission as measured by PCDAI <10 points after induction therapy (four of 11 used IFX as induction therapy, and seven of 11 received EEN as induction therapy). Three of 27 did not acquire clinical remission, and 13 of 27 did not complete their induction therapy. The demographic and clinical features of patients with CD before and after therapy were summarized in Table 1. A total of 11 age- and gender-matched Ctr were enrolled in the study. The clinical parameters of CD patients and controls were summarized in Table 2. There was no significant difference in age, gender, height for age (HFA) *z*-score, and body mass index (BMI) *z*-score between patients with CD and Ctr. The levels of inflammatory indicator such as white blood cell (WBC) count and neutrophil (NE) count were significantly increased in the patients with CD ($P < 0.05$). Other laboratory markers including hemoglobin (HB) and aspartate transaminase (AST) were statistically lower, while blood platelet (Plt) was higher in the patients with CD. The clinical parameters of aCD and aCDs groups were summarized in Table S1. There was no significant difference in age, gender, HFA *z*-score, BMI *z*-score, WBC, NE, lymphocyte (LY), red blood cell (RBC), HB, Plt, TBA, alanine aminotransferase (ALT), AST, and creatinine (Cre) between the two groups.

### Induction therapy improved the gut dysbiosis of CD patients

The GM was initially compared between aCD patients and Ctr, and between aCD patients and them in remission. *Firmicutes* was the most dominant phyla in the Ctr group and rCD group; however, *Proteobacteria* was the most dominant one in the aCD group and aCDs group (Fig. S1A and B). *Bifidobacterium* was the most dominant bacteria in patients with Ctr, and *Mediterraneibacter* was the most dominant one in rCD group. In comparison, *Escherichia*, *Enterococcus*, and *Klebsiella* were the three dominant genera in the aCD group (Fig. 2A and B). We observed higher α-diversity in the gut microbiota of Ctr

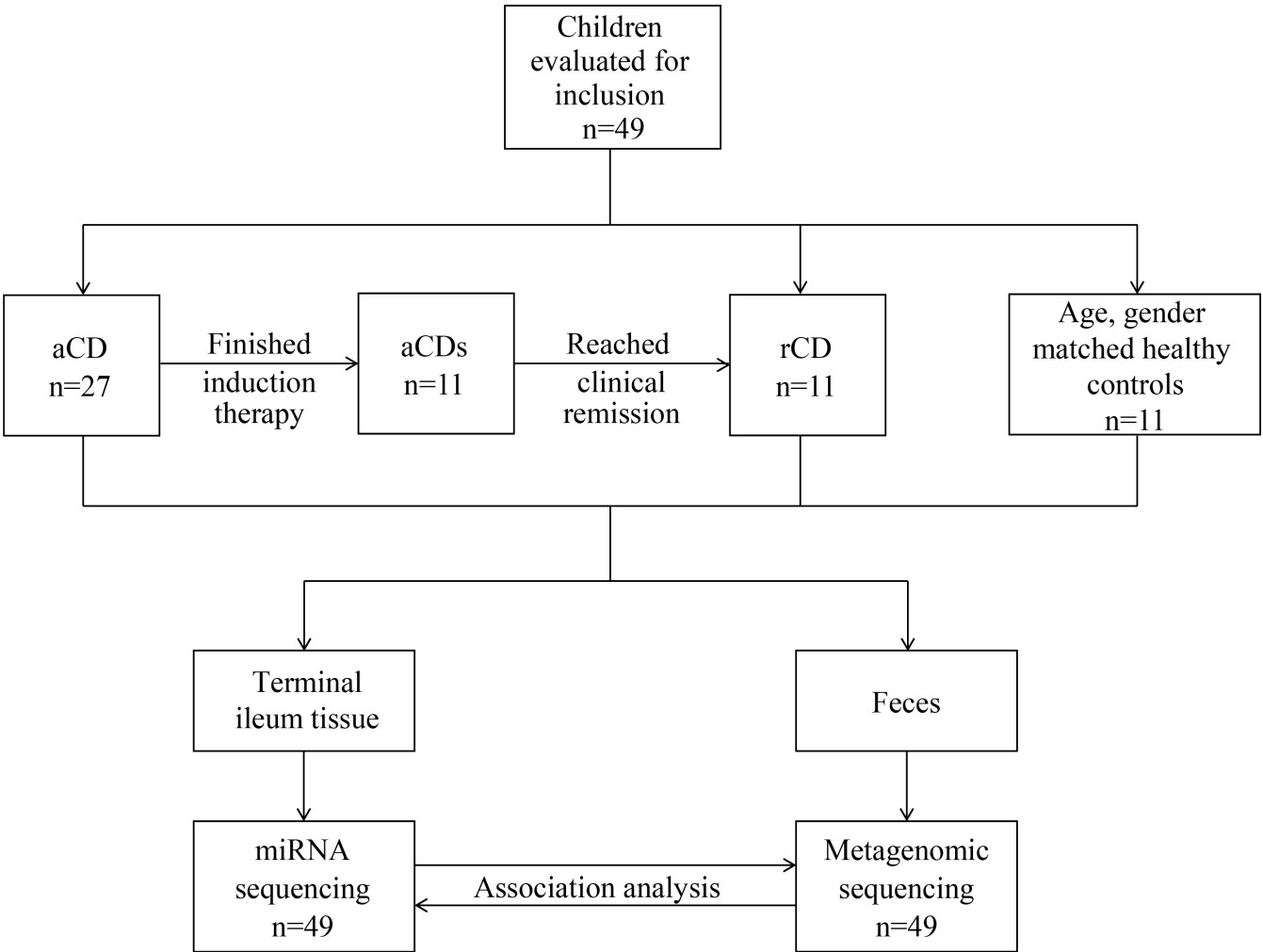

**FIG 1** Experimental design workflow: patient recruitment and sample collection and their examination.

compared with aCD group and rCD group versus aCDs group at species levels based on Shannon diversity index although there was no significant difference at the phylum level (Fig. 2C and E; Fig. S1C and E). The PCoA of beta diversity showed that the GM structure of all the two groups were clearly separated at both species and phylum levels (Fig. 2D and F; Fig. S1D and F).

## Seven key species were significantly changed in CD patients

The significant differences in gut microbes among the four groups are shown in Fig. 3A and B. There were 392 microbes increased in aCD group compared with Ctr group, and 2,845 GM in species level were found lower in aCD group (Fig. 3A). A total of 758 bacteria in species level were higher in rCD group compared with aCDs group, and 50 microbes were lower (Fig. 3B). Among which, we identified a total of 109 differential microbes that met the following screening criteria: FC (aCD/Ctr < 2 and rCD/aCDs > 2), $P < 0.01$ (Table S2). To further evaluate whether altered GM profiles were associated with the disease severity, we examined their correlations with PCDAI, CDEIS score, and calprotectin. We found that 156 bacterial species were closely related with PCDAI, CDEIS score, and calprotectin (Table S3). Among them, *D. raffinosedens*, *T. metallivorans*, *R. intestinalis*, and *Dorea* sp. AGR2135 were overlapped in both Tables S1 and S2 (Fig. 3D). As shown in Table S2, FC (aCD/Ctr) of *D. raffinosedens*, *T. metallivorans*, *R. intestinalis*, and *Dorea* sp. AGR2135 were 0.41038, 0.07634, 0.13398, and 0.08822, and FC (rCD/aCDs) of *D. raffinosedens*, *T.*

**TABLE 1** Clinical features of pediatric patients with CD[a]

| Patient ID | Age | Gender | Disease location (Paris L) | PCDAI | | CDEIS | | Calprotectin (µg/g) | |
|---|---|---|---|---|---|---|---|---|---|
| | | | | aCDs | rCD | aCDs | rCD | aCDs | rCD |
| 1 | 14 yrs 9 mos | M | A1b, L3 + L4b, B1, p, G0 | 15 | 5 | 7.2 | 0 | 60.88 | 63.881 |
| 2 | 12 yrs 6 mos | M | A1b, L3 + L4b, B1, G0 | 22.5 | 0 | 10.8 | 0 | 365.23 | 144.36 |
| 3 | 15 yrs 7 mos | M | A1b, L3 + L4b, B1, G0 | 17.5 | 5 | 18.4 | 2.2 | 267.42 | 36.85 |
| 4 | 12 yrs 9 mos | F | A1b, L3 + L4b, B1, G0 | 37.5 | 0 | 12.4 | 1.9 | 445.14 | 139.68 |
| 5 | 11 yrs 4 mos | M | A1b, L3 + L4a + L4b, B1, G0 | 17.5 | 0 | 21.6 | 0.2 | 570.22 | 33.58 |
| 6 | 12 yrs 10 mos | M | A1b, L3, B1, G1 | 25 | 0 | 17.6 | 9.6 | 696.24 | 305.33 |
| 7 | 13 yrs 8 mos | M | A1b, L1 + L4b, B1, G0 | 20 | 5 | 2.44 | 1.3 | 63.52 | 264.88 |
| 8 | 13 yrs 4 mos | M | A1b, L3 + L4b, B1, G0 | 40 | 5 | 25 | 4.2 | 58.43 | 342.48 |
| 9 | 13 yrs 7 mos | M | A1b, L3 + L4a + L4b, B1, P, G0 | 22.5 | 0 | 5.2 | 0 | 275.16 | 136.08 |
| 10 | 13 yrs | M | A1b, L3 + L4b, B1, P, G1 | 42.5 | 0 | 12.6 | 0 | 164.58 | 30.65 |
| 11 | 11 yrs 11 mos | M | A1a, L3 + L4a + L4b, B1, p, G0 | 12.5 | 5 | 1.24 | 0 | 32.87 | 32.71 |

[a]BMI: body mass index; CDEIS: Crohn's Disease Endoscopic Index of Severity; CRP: C reactive protein; HFA: height for age; PCDAI: Pediatric Crohn's Disease Activity Index; M: male; F: female.

*metallivorans*, *R. intestinalis*, and *Dorea* sp. AGR2135 were 57.33034, 17.42137, 8.78935, and 15.26307. The top 20 most abundant microbial of each microbial state at the genus level for each group were shown in Fig. 3D. There were 11 bacteria both meeting the following standards: FC (aCD/Ctr > 2 and rCD/aCDs < 2, $P$ < 0.05) (Table S4). Among them, *E. coli*, *S. sonnei*, and *S. enterica* were also listed in Fig. 3D.

## The proportions of seven key gut microbes were changed among groups

As shown in Fig. 4A through G, the proportions of *D. raffinosedens*, *T. metallivorans*, *R. intestinalis*, and *Dorea* sp. AGR2135 were reduced in aCD group compared with Ctr group and then increased in rCD group after remission. Meanwhile, the proportion of *E. coli*, *S. sonnei*, and *S. enterica* were increased in patients with CD and then decreased after induction therapy. These results from metagenomic sequencing (Fig. 2) were consistent with PCR data.

**TABLE 2** Comparison of the pediatric patients with CD and the Ctr[a]

| Parameter | Ctr | CD | $P$ value |
|---|---|---|---|
| $n$ | 11 | 27 | |
| Age (yr) | 12.16 ± 0.90 | 13.09 ± 1.46 | 0.06 |
| Males/females | 8/3 | 18/9 | 1.00 |
| HFA $z$-score | 0.45 ± 1.22 | 0.14 ± 1.16 | 0.47 |
| BMI $z$-score | −0.36 ± 1.54 | −1.51 ± 1.87 | 0.08 |
| WBC ($10^9$/L) | 5.83 ± 1.06 | 9.72 ± 3.79 | **<0.01**[b] |
| NE ($10^9$/L) | 2.91 ± 0.78 | 6.55 ± 2.93 | **<0.01** |
| LY ($10^9$/L) | 2.35 ± 0.69 | 2.14 ± 1.09 | 0.56 |
| RBC ($10^{12}$/L) | 4.39 ± 0.36 | 4.37 ± 0.64 | 0.93 |
| HB (g/L) | 124.18 ± 15.17 | 101.26 ± 27.95 | **0.02** |
| Plt ($10^9$/L) | 286.45 ± 91.12 | 455.04 ± 110.37 | **<0.01** |
| TBA (µmol/L) | 5.53 ± 6.60 | 6.01 ± 4.82 | 0.80 |
| ALT | 16.18 ± 6.01 | 11.11 ± 7.95 | 0.07 |
| AST | 32.09 ± 8.13 | 17.19 ± 7.31 | **<0.01** |
| Cre | 56.82 ± 12.77 | 53.48 ± 16.53 | 0.55 |

[a]ALT: alanine transaminase; AST: aspartate transaminase; BMI: body mass index; CD: Crohn's disease; Cre: creatinine; Ctr: non-IBD controls; HB: hemoglobin; HFA: height for age; LY: lymphocyte; NE: neutrophil; Plt: blood platelet; RBC: red blood cell; TBA: total biliary acid; WBC: white blood cell.
[b]Bold font is used to highlight the parameters where $P$ < 0.05.

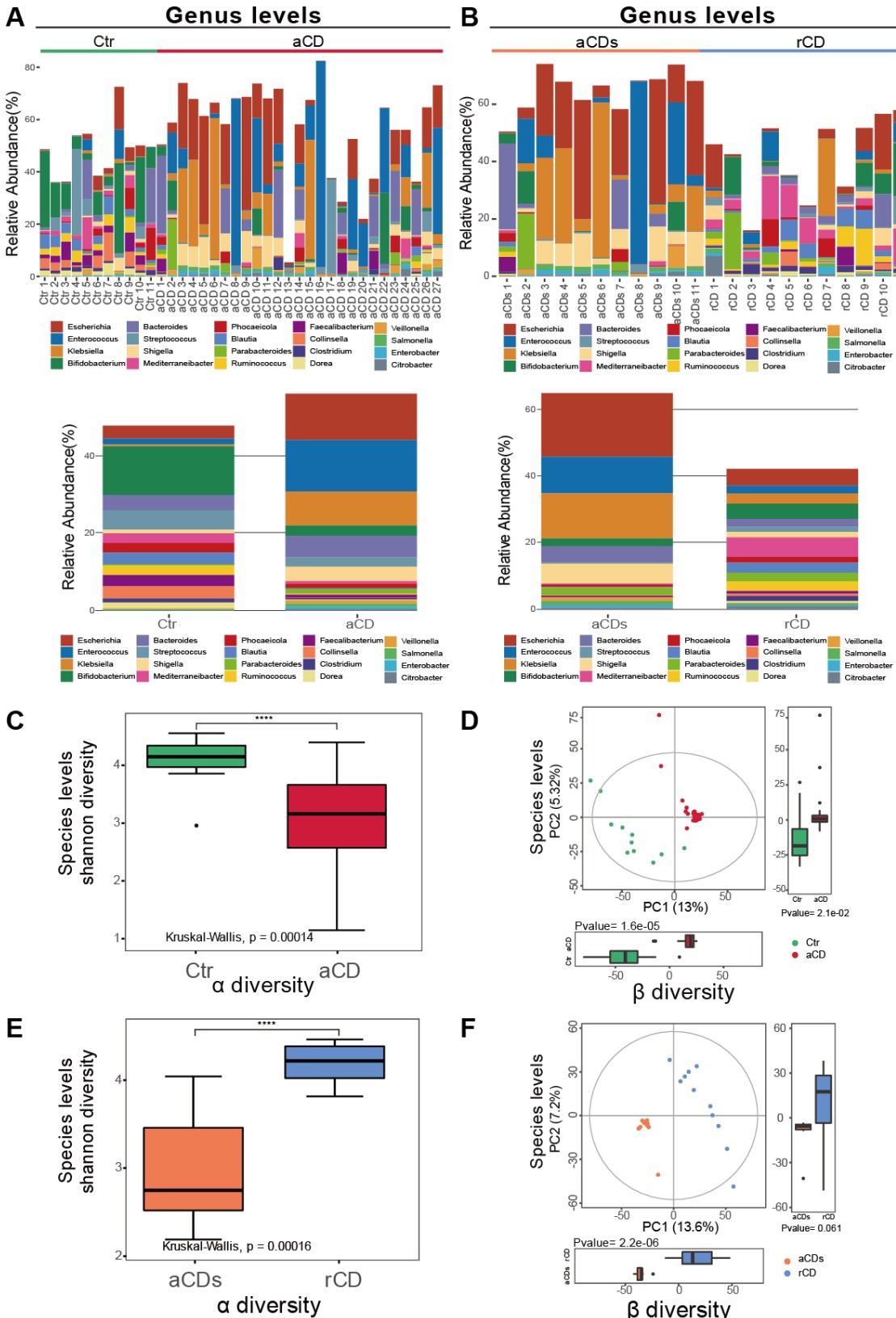

**FIG 2** Differential GM profiles in four groups. Relative abundance of the 20 most abundant microbial and mean community composition of each microbial state at the genus level (A and B) for each group. (C and E) Shannon diversity differs significantly across microbial states at the species level for each group. Kruskal–Wallis for species levels, $P_{(Ctr\ vs\ aCD)}$ = 0.00014, $P_{(rCD\ vs\ aCDs)}$ = 0.00016. (D and F) PCoA profile of microbial diversity illustrates that gut bacterial communities are compositionally distinct at the species level for each group.

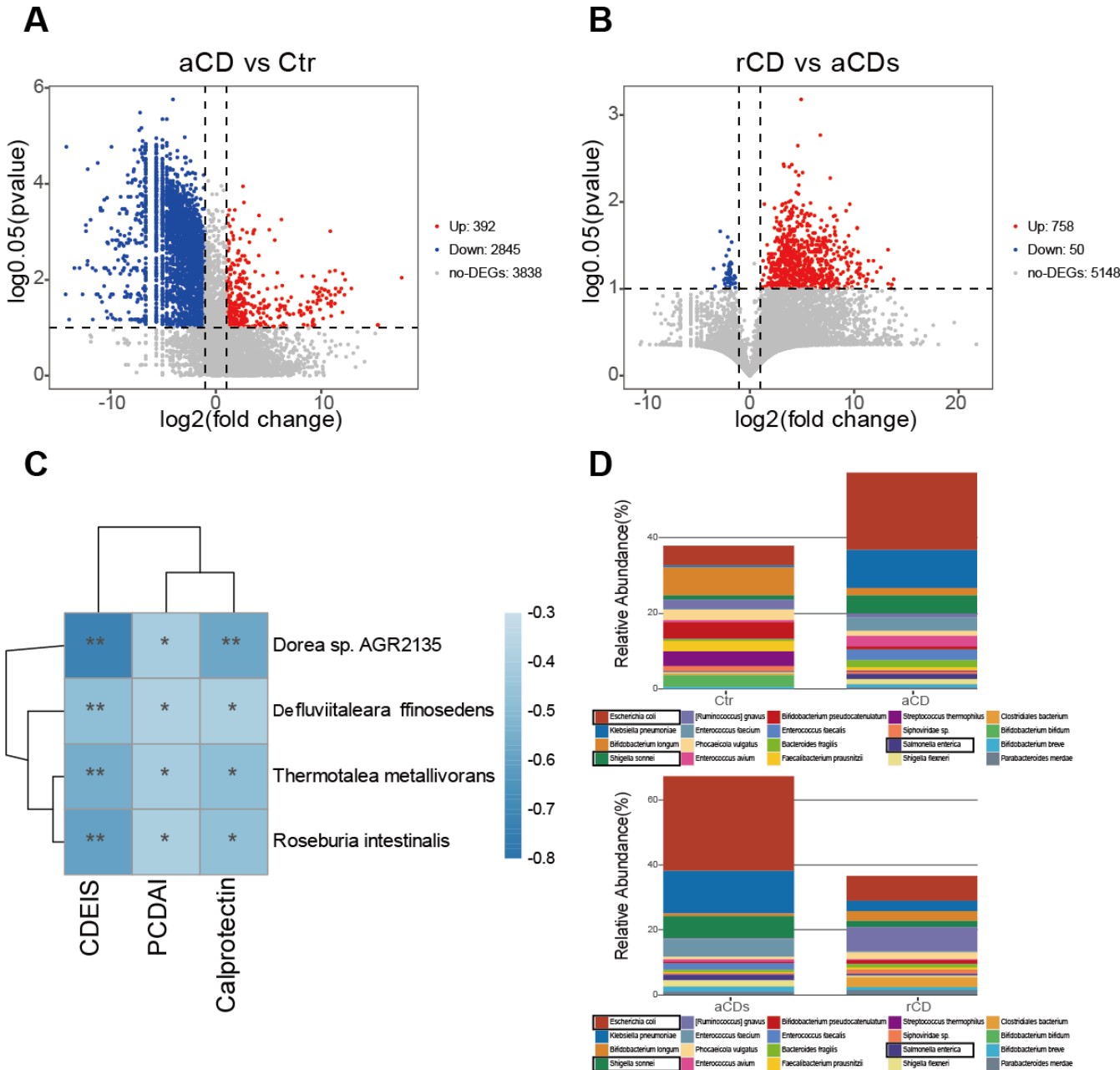

**FIG 3** Key gut microbes among four groups. (A and B) Volcano plots indicating the variation of fecal GM between the two groups (FC (CD/Ctr, rCD/aCDs) > 2 or FC (CD/Ctr, rCD/aCDs) < 0.5, *P* < 0.05). (C) Heat map of Spearman's correlations between GM and PCDAI, CDEIS score, and calprotectin. *: *P* < 0.05. (D)Top 20 differential fecal microbiota at species level.

## The miRNA profiles were changed in CD patients

The terminal ileum tissue miRNA profiles of patients with CD were analyzed pre- and post-induction therapy and compared with Ctr. The top 20 most abundant miRNAs of the four groups were listed in Fig. 5A. Volcano plot showed that there were 63 miRNAs upregulated and 37 miRNAs downregulated in the aCD group compared with Ctr (FC [aCD/Ctr < 0.8 or aCD/Ctr > 1.2] and *P* < 0.05 [Fig. 5B]). Additionally, two miRNAs were upregulated and 174 miRNAs downregulated in the rCD group compared with aCDs group (FC [rCD/aCDs > 1.2 or rCD/aCDs < 0.8] and *P* < 0.05 [Fig. 5B]). Among these, there were 37 miRNAs meeting the following standard: FC (aCD/Ctr < 0.8 and rCD/aCDs > 1.2),

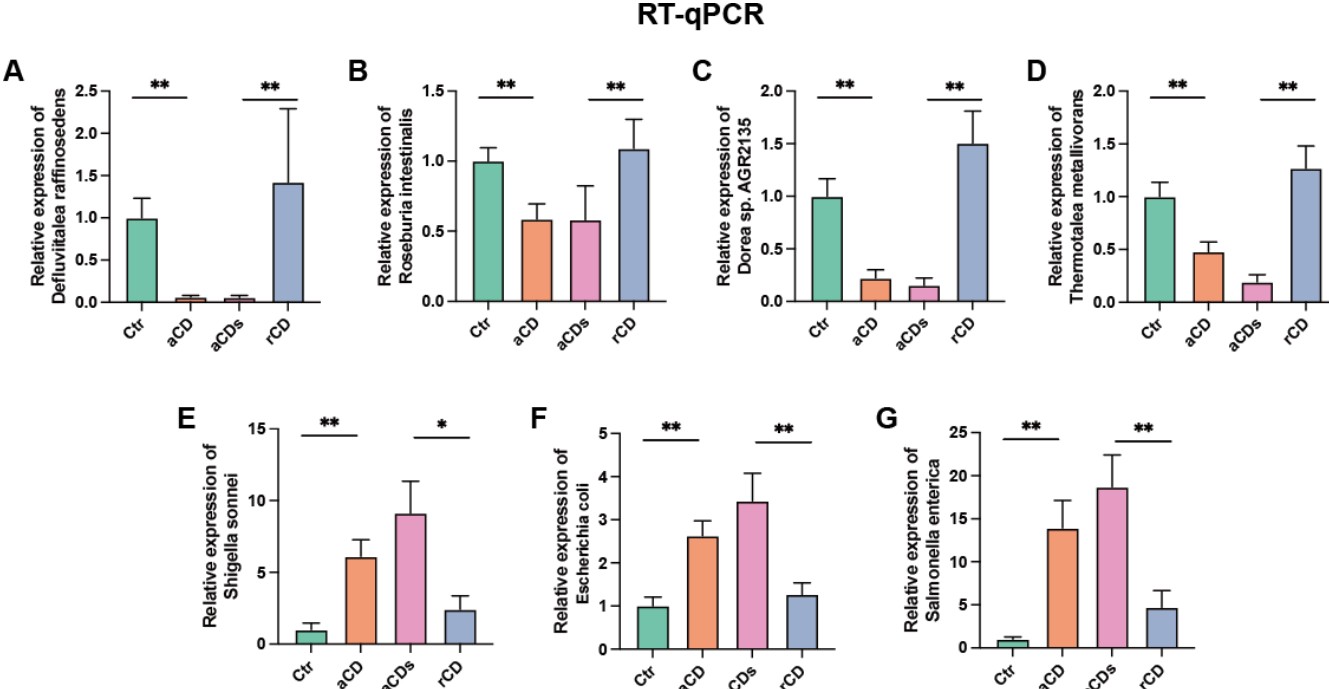

**FIG 4** Relative proportion of *D. raffinosedens*, *T. metallivorans*, *R. intestinalis*, *Dorea* sp. AGR2135, *E. coli*, *S. sonnei*, and *S. enterica* among groups. (A–G) The relative proportion of *D. raffinosedens*, *T. metallivorans*, *R. intestinalis*, *Dorea* sp. AGR2135, *E. coli*, *S. sonnei*, and *S. enterica* in four groups from RT-qPCR data. *: *P* < 0.05, **: *P* < 0.01.

*P* < 0.05; FC (aCD/Ctr > 1.2 and rCD/aCDs < 0.8), and *P* < 0.05. (Fig. 5C). Among them, two miRNAs hsa-miR-215-5p and hsa-miR-194-5p were also showed in Fig. 5A. Their relative expression among four groups were listed in Fig. 5D and E.

## Differentially expressed miRNAs and their functional analysis

In order to further estimate whether altered miRNA profiles were associated with CD disease states, terminal ileum mucosal states, and terminal ileum inflammation, we investigated if miRNA profiles were related to PCDAI, CDEIS score, and calprotectin. We found that the expressions of two miRNAs hsa-miR-12135 and hsa-miR-509-3-5p were negatively related to the PCDAI, CDEIS score, and calprotectin. On the contrary, the expression of four miRNAs hsa-miR-212-5p, hsa-miR-4448, hsa-miR-501-3p, and hsa-miR-503-5p had positive correlations with both PCDAI, CDEIS score, and calprotectin (Fig. 6A). Their correlation coefficients with PCDAI, CDEIS score, and calprotectin were shown in Table S5. Then, functional enrichment analysis was performed on eight key miRNAs to explore the potential mechanisms regulating the pathogenesis of CD, including hsa-miR-215-5p, hsa-miR-194-5p, hsa-miR-12135, hsa-miR-509-3-5p, hsa-miR-212-5p, hsa-miR-4448, hsa-miR-501-3p, and hsa-miR-503-5p. The top 10 significantly enriched GO functions include cellular component, biological process, and molecular function in Fig. 6B, including cyclin-dependent protein kinase holoenzyme complex, regulation of cyclin-dependent protein serine/threonine kinase activity, cyclin binding, and cyclin-dependent protein serine/threonine kinase regulator activity. As shown in Fig. 6C, the top 10 significantly enriched KEGG pathways mainly included cell cycle, p53 signaling pathway, and HIF-1 signaling pathway.

## Correlations between differential bacteria and miRNAs

To study the potential dependencies between microbiome and miRNA, the correlations between the two datasets were examined. The correlation Sankey chart analysis visually

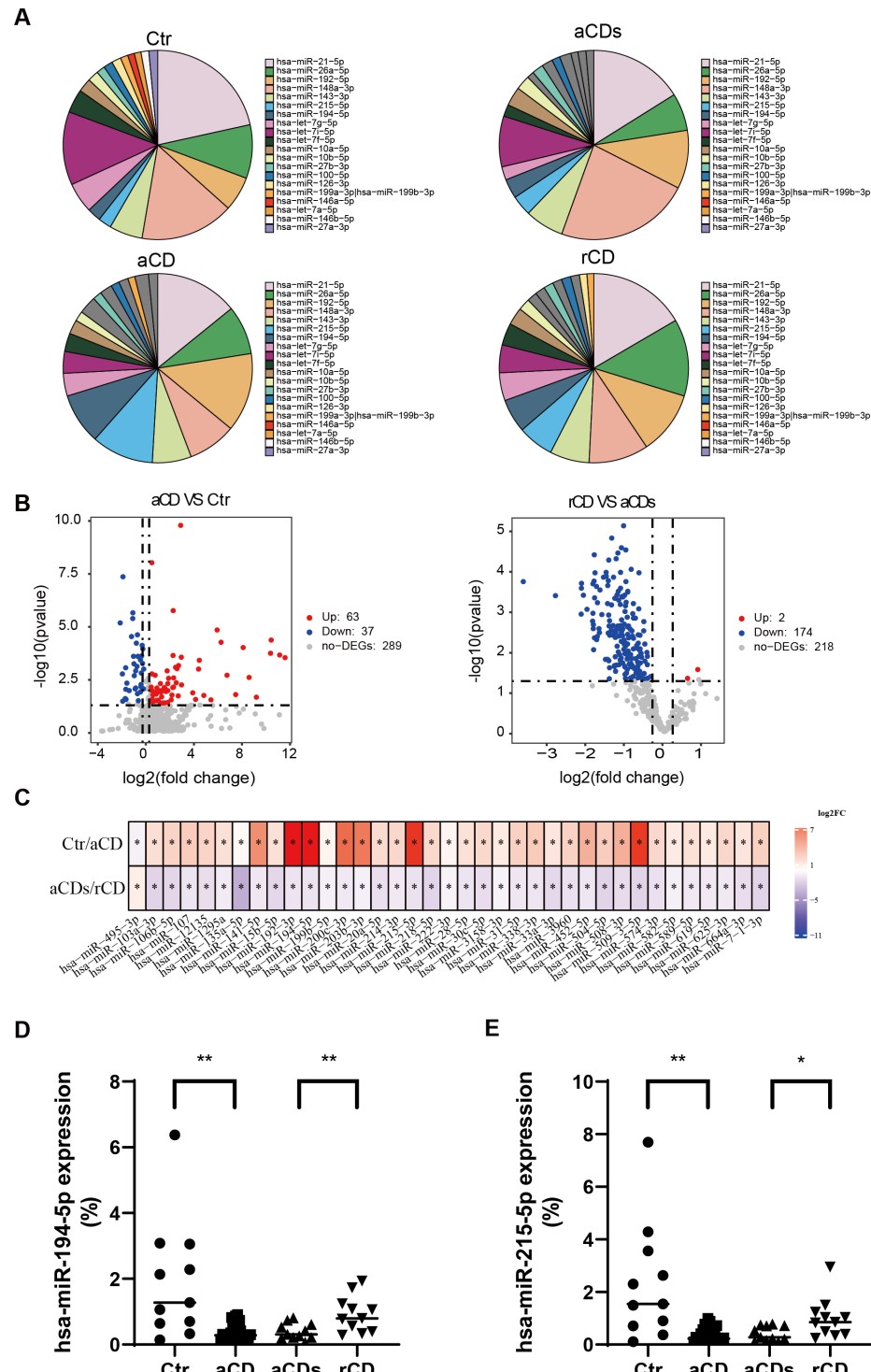

**FIG 5** Differential miRNA profiles in four groups and two key miRNAs were found. (A) Relative abundance of the 20 most abundant miRNAs in each group. (B) Volcano plot of differentially expressed miRNAs between Ctr and aCD groups (FC [CD/Ctr] > 1.2 or FC [CD/Ctr] <0.8, $P < 0.05$). (C) Volcano plot of differentially expressed miRNAs between rCD and aCDs groups (FC [rCD/aCDs] > 1.2 or FC [rCD/aCDs] < 0.8, $P < 0.05$). (D) Colon tissue miRNAs expressed differentially in patients with CD compared with Ctr according to the following standard: FC (aCD/Ctr < 1 and rCD/aCDs > 1), $P < 0.05$; FC (aCD/Ctr > 1 and rCD/aCDs < 1), $P < 0.05$. (D and E) The relative expression of hsa-miR-215–5p and hsa-miR-194–5p in each group. *: $P < 0.05$, **: $P < 0.01$.

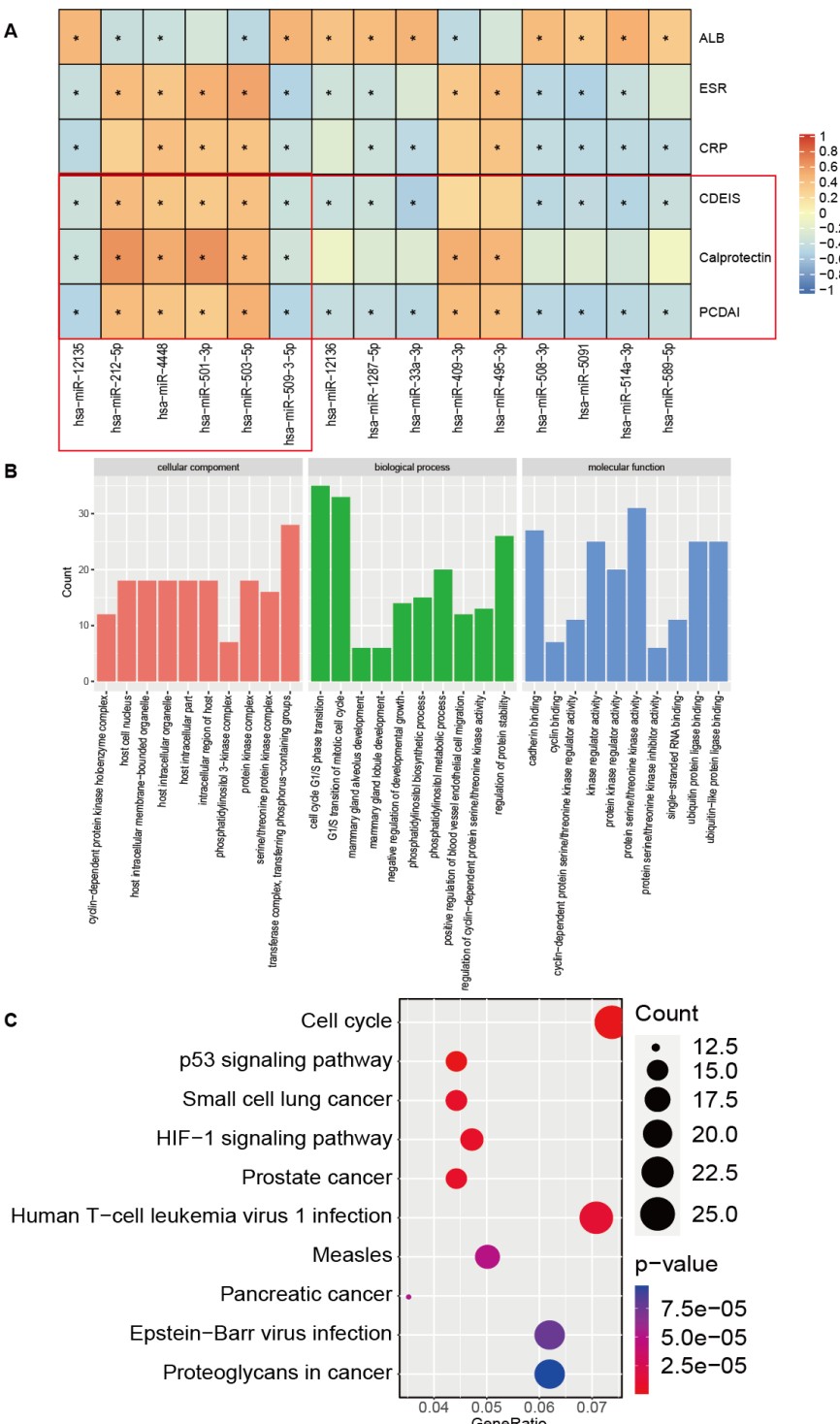

**FIG 6** Correlation analysis between miRNAs and key clinical indicators and functional enrichment analysis based on the eight different expressions of miRNAs. (A) Heat map of Spearman's correlations between miRNAs and clinical data such as PCDAI, CDEIS score, calprotectin, erythrocyte sedimentation rate (ESR), C-reactive protein (CRP), and albumin (ALB). *: $P < 0.05$. (B) The top 10 significantly enriched GO terms by eight key miRNAs analysis. horizontal axis: title of GO terms; vertical axis: number of mRNAs. (C) Results of top 10 significantly enriched KEGG pathway analysis of the differentially expressed genes (DEGs). The circle size is proportional to the number of DE mRNAs in each biological process. The color of the circle is correlated with the $P$ value, going from blue to red with decreasing $P$ value.

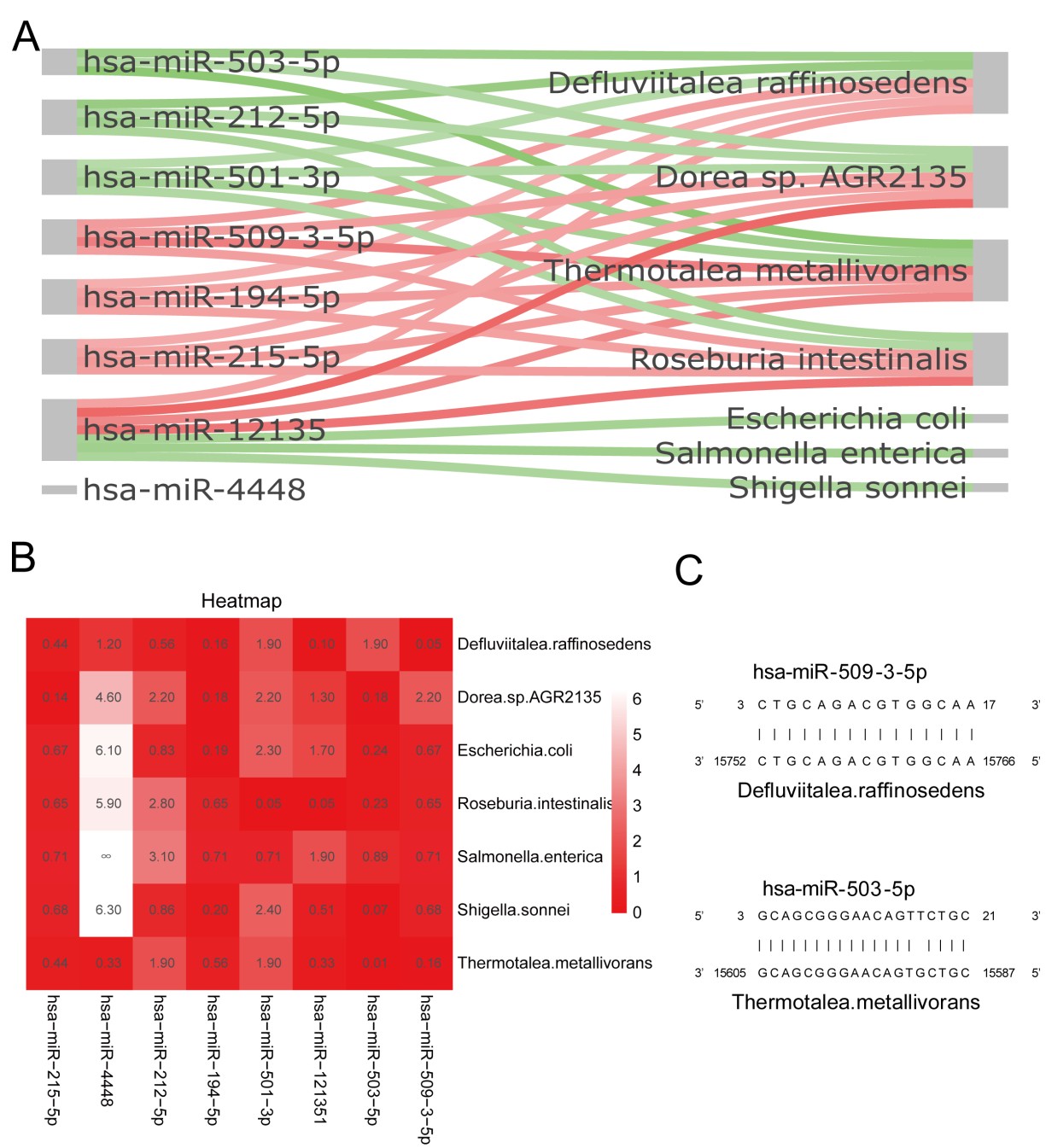

**FIG 7** Correlation analysis between the expression of eight key miRNAs and seven key gut microbes. (A) Correlation Sankey diagram analysis between the hsa-miR-215-5p, hsa-miR-194-5p, hsa-miR-12135, hsa-miR-509-3-5p, hsa-miR-212-5p, hsa-miR-4448, hsa-miR-501-3p, hsa-miR-503-5p, and *D. raffinosedens*, *T. metallivorans*, *R. intestinalis*, *Dorea* sp. AGR2135, *E. coli*, *S. sonnei*, and *S. enterica*. Red line means positive correlation while green negative correlation. *P* < 0.05. (B) Heat map showing minimal *e*-value of each pair between miRNAs and gut microbes. The *e*-value is smaller; the red goes deeper. (C) Schematic diagram of the putative binding sites of the seed sequence of hsa-miR-509-3-5p and hsa-miR-503-5p in the genes of *D. raffinosedens* and *T. metallivorans*.

demonstrated the association between seven key gut microbiota (*D. raffinosedens*, *T. metallivorans*, *R. intestinalis*, *Dorea* sp. AGR2135, *E. coli*, *S. sonnei*, and *S. enterica*) and eight key miRNAs (hsa-miR-215-5p, hsa-miR-194-5p, hsa-miR-12135, hsa-miR-509-3-5p, hsa-miR-212-5p, hsa-miR-4448, hsa-miR-501-3p, and hsa-miR-503-5p) (Fig. 7A). hsa-miR-509-3-5p, hsa-miR-194-5p, hsa-miR-215-5p, and hsa-miR-12135 showed positive correlations with *D. raffinosedens*, *T. metallivorans*, *R. intestinalis*, and *Dorea* sp. AGR2135, while hsa-miR-503-5p, hsa-miR-212-5p, hsa-miR-501-3p, and hsa-miR-12135 were

negatively corrected with all the seven key GMs. The identified eight miRNAs and their target genes in the seven key gut microbes were predicted. The minimal *e*-value of each pair was shown in Fig. 7B. The putative binding sites of the seed sequence of selected and eight miRNAs in the genes of seven key gut microbes were listed in Fig. 7C and Fig. S3.

## DISCUSSION

Recent studies have highlighted the key role of gut microbiota in nutrition, immune system, and defense of the host (36–38). Increasing clinical and experimental data suggests the gut microbiota as a crucial actor in CD (39). Previous studies indicated that the α-diversity and beta diversity in gut microbiota were decreased in patients with CD when compared with healthy controls (40). This was consistent with our study in Fig. 2. Furthermore, gut dysbiosis with more harmful bacteria and less benign bacteria was found in CD. In our study, *D. raffinosedens*, *T. metallivorans*, *R. intestinalis*, and *Dorea* sp. AGR2135 were decreased in patients with aCD and increased in in patients with rCD (Fig. 3C), which suggested their potential probiotic effects for CD, while the other three bacteria such as *E. coli*, *S. sonnei*, and *S. enterica* may be harmful to CD for the reason that they were increased in aCD groups and decreased in rCD groups (Fig. 3D).

One of the main modes that the gut microbiota interacts with the host is by means of miRNAs (41). Interactions between the host miRNAs and gut microbiota are crucial for shaping gut homeostasis and immune response in CD (42). MiRNAs are small non-coding RNAs capable of regulating gene expression post-transcriptionally (43). The growth of gut microbes and composition of intestinal microbiota can be altered by miRNAs secreted by intestinal epithelial cells (44). So far, few studies have explored the associations between the gut microbiota and tissue miRNAs and how the miRNAs regulate gut microbiota composition in CD. Our study attempted to investigate the relationship between miRNAs and gut microbiota in the different stages of CD and found out the underlying mechanism how miRNAs alter microbiome composition. First, we identified two key miRNAs hsa-miR-215 and hsa-miR-194-5p based on their differential expressions among different group comparisons (Fig. 5D and E) and six specific miRNAs (hsa-miR-12135, hsa-miR-509-3-5p, hsa-miR-212-5p, hsa-miR-4448, hsa-miR-501-3p, and hsa-miR-503-5p) that had close relationship with PCDAI, CDEIS score, and calprotectin (Fig. 6A). Then, we identified that seven key gut microbes and seven specific miRNAs had significantly positive or negative correlations with each other (Fig. 7A). It was consistent with our study that *E. coli* had a strong relationship with miR-509-3-5p (45) and miR-194a-5p acted as an important regulator in *Salmonella*-infected disease (46). Among patients with functional constipation, hsa-miR-215-5p was inversely associated with *UCG.002*, while positively interacted with *Lachnospiraceae_NK4A136_group* (47). At last, in order to find out the exact mechanism between miRNAs and gut microbiota, the eight key miRNAs and their target genes in the seven key gut microbes were predicted further (Fig. 7B). From our result, the most likely bacterial mRNA targets of a given miRNA were listed in Fig. 3, and most of those miRNAs could affect bacterial growth by target regulation of their genes.

The functional features of the eight key miRNAs like hsa-miR-215-5p, hsa-miR-194-5p, hsa-miR-12135, hsa-miR-509-3-5p, hsa-miR-212-5p, hsa-miR-4448, hsa-miR-501-3p, and hsa-miR-503-5p need to be further studied. The GO functions indicated that cyclin-dependent protein kinase holoenzyme complex, regulation of cyclin-dependent protein serine/threonine kinase activity, cyclin binding, and cyclin-dependent protein serine/threonine kinase regulator activity were significantly enriched (Fig. 6B). Cyclin and cyclin-dependent protein play an important role in IBD. Cyclin D1 is a cell cycle regulatory protein that is upregulated in IBD in both epithelial and immune cells (48). And the expression patterns of cyclin D1 and cyclin E in IBD may indicate their contribution in epithelial cell turnover (49). It was found that cyclin-dependent kinase 8 inhibition can upregulate interleukin-10 (IL-10) abundance and then the high level of IL-10 can suppress the progression of pathogenic inflammation such as IBD (50).

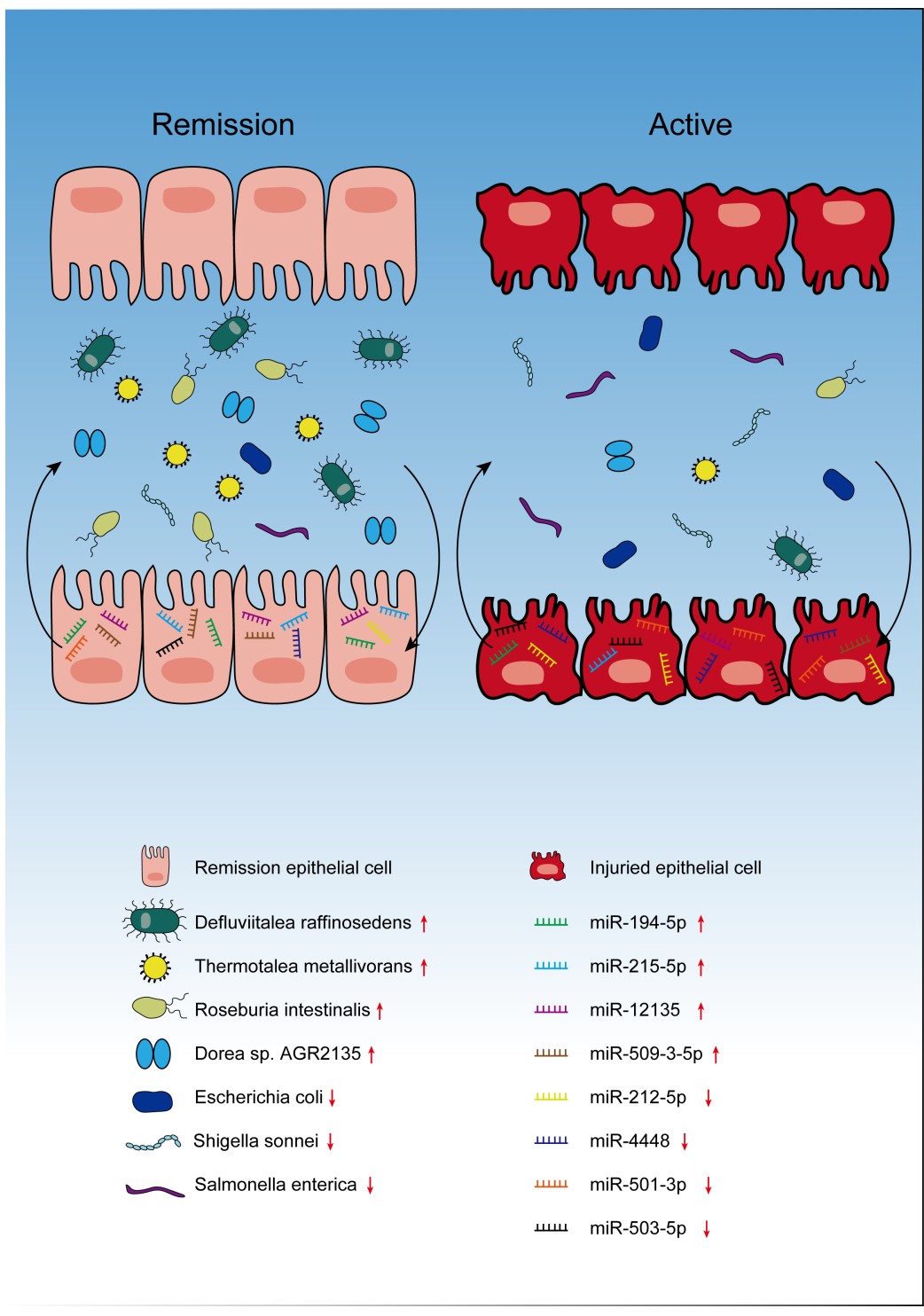

**FIG 8** Schematic illustration for the changes of seven key fecal GM and eight key tissue microRNA in different stages of pediatric CD.

Moreover, KEGG pathway enrichment analyses were performed based on predictive miRNAs and found that several pathways such as cell cycle and signaling pathways regulated were significantly enriched (Fig. 6C). The results indicated that miRNAs play crucial roles in the development and progression of CD. Furthermore, qPCR data from

total microbial loads of *D. raffinosedens*, *R. intestinalis*, *Dorea* sp. AGR2135, *T. metallivorans*, *E. coli*, *S. sonnei*, and *S. enterica* in groups further validated the microbial changes (Fig. 4).

To summarize, this study reported the close correlations between seven key gut microbes and eight specific miRNAs in the patients with active or rCD (Fig. 8). The key differential miRNAs and gut microbes were carefully selected by comparing the gut microbial and miRNA profiles at different stages of CD during clinical induction. However, their underlying mechanistic functions need to be further explored in the future. Furthermore, current study is limited by the single-center cohort with relatively small sample size. More independent studies or multicenter cohort studies focusing on the potential cross-talk between GM and intestinal miRNAs are required for further validation.

## Conclusions

This study reported for the first time the close relationship between the seven key gut microbes and eight specific miRNAs in the patients with active or rCD. Our findings are expected to provide potential biomarkers that might contribute to a better understanding of CD pathogenesis and treatment responses and offer the new therapeutic targets for CD management.

## ACKNOWLEDGMENTS

Thanks go to Xiaoli Shu for the technical support of the fecal calprotectin detection experiment. We thank Genefund Biotech (Shanghai, China) for the assistance in the data analysis.

This work was supported partly by Zhejiang Provincial National key research and development program (2019C03037) and the Natural Science Foundation of Zhejiang Province (LQ22H160006). Yan Ni is supported by the National Natural Science Foundation of China (82170583 and 81900510), and the National Key Research and Development Program of China (2021YFC2701904).

Conceived and designed the experiments: Y.N., J.C., and Y. Lv. Participant enrollment, sample collection and processing: Y. Lv, Y.H., A.L., Y. Lou, and Q.C. Wrote the manuscript: Y. Lv, Y.N., C.Z., Y.H., Y. Lou, and J.C. EEN and IFX therapy and clinical follow-up: Y. Lv, J.C., Q.C., G.Y., Y. Lou, J.L., J.Y., Y.F., H.Z., and K.P. C.X. performed the statistical analysis of metagenomic and metabolomic data. All authors have read and approved the final manuscript.

## AUTHOR AFFILIATION

[1]Gastroenterology Department, Children's Hospital Zhejiang University School of Medicine, National Clinical Research Center for Child Health, Hangzhou, China

## AUTHOR ORCIDs

Changjun Zhen  http://orcid.org/0009-0007-7412-2245
Jie Chen  http://orcid.org/0000-0002-5929-7262
Yan Ni  http://orcid.org/0000-0003-1779-7266

## DATA AVAILABILITY

The raw sequence data reported in this paper have been deposited in the Genome Sequence Archive (51) in National Genomics Data Center (52), China National Center for Bioinformation/Beijing Institute of Genomics, Chinese Academy of Sciences (GSA-Human: HRA007915 and GSA-Human: HRA008007), publicly accessible at https://ngdc.cncb.ac.cn/gsa-human.

## ETHICS APPROVAL

The study was approved by the Medical Ethical Committee of Children's Hospital, Zhejiang University School of Medicine under 2019-IRB-109 and 2021-IRB-314. Written consent was completed and signed by the children and their legal guardians before samples were collected.

## ADDITIONAL FILES

The following material is available online.

### Supplemental Material

**Supplemental File (mSystems00783-24-s0001.docx).** Figures S1 to S3, Table S1, and Table S5.
**Table S2 (mSystems00783-24-s0002.xlsx).** The 109 different bacteria meeting the following standard: FC (aCD/Ctr < 2 and rCD/aCDs > 2), $P < 0.01$.
**Table S3 (mSystems00783-24-s0003.xlsx).** The 156 bacterial species which had relationship with both PCDAI, CDEIS score and calprotectin, $P < 0.05$.
**Table S4 (mSystems00783-24-s0004.xlsx).** The 11 bacteria meeting the following standard: FC (aCD/Ctr > 2 and rCD/aCDs < 2), $P < 0.05$.

### Open Peer Review

**PEER REVIEW HISTORY (review-history.pdf).** An accounting of the reviewer comments and feedback.

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
