## [Reviewer comments · mSystems]

Profiles and interactions of gut microbiome and intestinal microRNAs in pediatric Crohn's disease

Yao Lv, Changjun Zhen, Ana Liu, Yudie hu, Gan Yang, Cuifang Xu, Yue Lou, Qi Cheng, Youyou Luo, Jindan Yu, Youhong Fang, Hong Zhao, Kerong Peng, Yu Yu, Jingan Lou, Jie Chen, and Yan Ni

Corresponding Author(s): Yan Ni, The Children's Hospital, Zhejiang University School of Medicine, National Clinical Research Center for Child Health

Review Timeline:

Submission Date:

June 13, 2024

Accepted:

June 26, 2024

Editor: Promi Das

Reviewer(s): Disclosure of reviewer identity is with reference to reviewer comments included in decision letter(s). The following individuals involved in review of your submission have agreed to reveal their identity: Peng Xu (Reviewer #1); Zhao Yicheng (Reviewer #2); Susanta Chatterjee (Reviewer #3)

Transaction Report:

DOI: <https://doi.org/10.1128/msystems.00783-24>

Re: mSystems00783-24 (Profiles and interactions of gut microbiome and intestinal microRNAs in pediatric Crohn's disease)

Dear Dr. Yan Ni:

Your manuscript has been accepted, and I am forwarding it to the ASM production staff for publication. Your paper will first be checked to make sure all elements meet the technical requirements. ASM staff will contact you if anything needs to be revised before copyediting and production can begin. Otherwise, you will be notified when your proofs are ready to be viewed.

Sincerely,
Promi Das